# Language Models with Rationality

**Nora Kassner[1,2]    Oyvind Tafjord[1]    Ashish Sabharwal[1]    Kyle Richardson[1]**
**Hinrich Schütze[2]    Peter Clark[1]**

[1]Allen Institute for AI, Seattle, WA
[2]Center for Information and Language Processing, LMU Munich, Germany
kassner@cis.lmu.de
{oyvindt,ashishs,kyler,peterc}@allenai.org

## Abstract

While large language models (LLMs) are proficient at question-answering (QA), it is not always clear how (or even if) an answer follows from their latent "beliefs". This lack of interpretability is a growing impediment to widespread use of LLMs. To address this, our goals are to make model beliefs and their inferential relationships explicit, and to resolve inconsistencies that may exist, so that answers are supported by interpretable chains of reasoning drawn from a consistent network of beliefs. Our approach, which we call REFLEX, is to add a **rational, self-reflecting layer** on top of the LLM. First, given a question, we construct a **belief graph** using a backward-chaining process to materialize relevant model beliefs (including beliefs about answer candidates) and their inferential relationships. Second, we identify and minimize contradictions in that graph using a formal constraint reasoner. We find that REFLEX significantly improves consistency (by 8%-11% absolute) without harming overall answer accuracy, resulting in answers supported by faithful chains of reasoning drawn from a more consistent belief system. This suggests a new style of system architecture in which an LLM extended with a rational layer can provide an interpretable window into system beliefs, add a systematic reasoning capability, and repair latent inconsistencies present in the LLM.

## 1 Introduction

While large language models (LLMs) are impressive at question-answering (QA), it is not always clear how (or even if) an answer follows from their latent "beliefs"[1] about the world, or whether the LLM even has a coherent internal belief system. This general opacity is a growing impediment to widespread use of LLMs, e.g., in critical applications such as medicine, law, and hiring decisions,

---

[1] We adopt a simple definition of belief, namely that a model believes X if it answers "yes" to the question "Is X true?". Other definitions could also be used; see Section 2.

**Question:**
Which animal gives live birth? (A) giraffe (B) spider

**Direct model answers:**
S1: giraffes give live birth? → true
S2: spiders give live birth? → true

**"Rational" Layer:**

**LLM Layer:**

**(a) Generate Belief Graph**    **(b) Resolve Inconsistencies**

| | |
|---|---|
| S1: giraffes give live birth | S6: a spider is a rodent |
| S2: spiders give live birth | S7: rodents are mammals |
| S3: mammals give live birth | R1: S3 & S4 → S1 |
| S4: a giraffe is a mammal | R2: S5 & S6 → S2 |
| S5: rodents give live birth | R3: S3 & S7 → S5 |

Figure 1: (**Top**) When queried about each answer option independently, the model incorrectly believes both are true, and is more confident in the wrong answer ($S_2$). (**Bottom**) REFLEX adds a "rational" layer above the LLM layer, in which a **belief graph** is constructed (by iteratively querying the LLM, up/down arrows), containing relevant model-believed facts (white/grey = believed T/F) and their inferential relationships. Inconsistencies are then identified (red) and minimized by a constraint reasoner that flips T/F labels on beliefs (green ✓/X), here resulting in the correct answer ($S_1$, green box) + explanation (graph) by the overall system (blue).

where properties of explainability, interpretability, and trust are paramount. Our goal is to help alleviate such opacity by constructing an explicit representation of system beliefs and their inferential relationships (including to answer candidates), so that answers are supported by interpretable chains of reasoning. These constructed **belief graphs**, e.g., Figures 1 and 2, form a **rational layer** above the LLM explaining how answers follow from beliefs, and provide a window into some of the latent contents of the model, potentially helping users

understand and trust model answers.

In addition, when we do this, we find such graphs expose latent inconsistencies in the model's beliefs. We show how such inconsistencies can be resolved using constraint satisfaction techniques. When we do this, the rational layer becomes not just a window onto the model, but an active reasoning component in its own right in a *larger, overall system*, comprising the (frozen) LLM plus rational layer (blue box, Figure 1). We show this results in a more consistent set of beliefs in the overall system, without harming overall answer accuracy (although some individual answers may change). The result is answers supported by faithful, system-believed chains of reasoning drawn from a consistent belief system.

Our approach, called REFLEX, introduces a **rational layer** consisting of two parts. First, to produce a belief graph, we recursively ask the LLM to explain why each candidate answer might be true, expressed as a set of sentences that entail the answer. This builds on earlier work on generating entailment-based and chain-of-thought explanations (Tafjord et al., 2022; Weir and Durme, 2022; Wei et al., 2022). We then add a self-verification step to check that the model itself believes those generations (i.e., that the model believes what it says), allowing us to identify sentences reflecting the model's own internal knowledge. For example, in Figure 1, when asked to explain S1 ("giraffes give live birth"), the model generates S7 ([because] "mammals give live birth") and S4 ([and] "a giraffe is a mammal"). Self-querying then checks if the model actually believes its generations ("Do mammals give live birth?"). The answer ("yes"/"no") assigns a true/false (T/F) value to each generation, indicated in Figure 1 by white/grey nodes. This procedure is then applied recursively to the generated, supporting sentences. The resulting network of model beliefs and their dependencies provides a a window into the model.

Second, we apply a formal constraint reasoner to this graph to resolve inconsistencies, by finding the optimal (minimal cost, Section 3.3) way of flipping T/F values. For example, on the left in Figure 1, S2 and S3 ("spiders do/don't give live birth") are in an XOR relationship (i.e., exactly one must be false), but both are believed as true (white) by the LLM - a latent contradiction within the LLM. Constraint reasoning then seeks to remove such inconsistencies, here flipping the belief

value on S2 from T to F (Figure 1, right), repairing the contradiction. This builds on earlier techniques (Kassner et al., 2021; Mitchell et al., 2022; Jung et al., 2022), though in a notably richer setting with over 350 nodes and 80 constraints per question, joint inference across answer candidates, and a variety of constraint types. The overall result is a fully autonomous, self-reflective system that is able to deliberate (and if necessary change) its answers, thereby resolving latent inconsistencies that would otherwise go unnoticed, and provide faithful explanations drawn from a consistent belief system.

We evaluate our implementation of REFLEX on three datasets: EntailmentBank (Dalvi et al., 2021), OBQA (Mihaylov et al., 2018), and QuaRTz (Tafjord et al., 2019). We find that REFLEX is able to construct belief graphs with significantly improved consistency (by 8%-11% absolute) without harming overall answer accuracy. In addition, answers are now supported by a more consistent, system-believed chain of reasoning, providing a window into the previously latent beliefs of the model. Our contributions are thus:

1. A **new style of system architecture** in which an LLM is extended with a **rational layer** in which an explicit representation of system beliefs and relationships is constructed and which can be reasoned over. This layer provides an **interpretable window** into system beliefs, adds a systematic reasoning capablity, and allows latent inconsistencies present in the LLM to be repaired.

2. An implementation of this architecture demonstrating that the **consistency of the overall system's network of beliefs can be significantly improved** without harming answer accuracy. Answers are now supported by explicit, interpretable chains of reasoning drawn from a more consistent network of beliefs.

## 2 Related Work

**Materializing a Model's Internal Knowledge:** It is now well recognized that LLMs contain extensive world knowledge (Petroni et al., 2019, 2020; Davison et al., 2019; Peters et al., 2019; Jiang et al., 2020; Roberts et al., 2020) that somehow enables them to perform well. Recent work has attempted to expose that knowledge in various ways, both to justify answers and improve performance, and our work falls into this genre. Standard explanation generation methods (Wiegreffe and Marasović, 2021) can produce compelling explanations, but

**Question:**

Heating ice (A) changes it's chemical make-up (B) will leave a puddle *[correct]* (C) makes it even colder

**Growing the Belief Graph:**

Figure 2: Given a question, each answer choice is first converted to a hypothesis statement (**A**). The belief graph is then constructed in stages, first generating rules that conclude the hypotheses (**B**), then backward-chaining to generate rules concluding the premises of those first rules, etc., and adding in negated versions of graph statements connected with the originals via XOR links (e.g., nodes 11 and 12), until the stopping criterion is met (**C**). Statements are then labeled with the model's belief in them (true/false), found via self-querying (white = believed true, grey = believed false). Finally, logical conflicts are identified (colored red), and constraint satisfaction techniques are used to resolve them. In this case, as there is strong evidence that node 2 is actually true (7 & 6 → 2, not(19) → 2), the solver finds that the minimum cost repair is to flip node 2's label from FALSE to TRUE. Here, node 2 ends up being selected as the final answer, thus correctly answering the original question.

with no guarantee that the generated sequence of tokens expresses the model's internal knowledge, nor entails the actual answer. Similarly, chain-of-thought (CoT) (Wei et al., 2022) and Least-to-Most (Zhou et al., 2023) prompting generate (in different ways) a step-by-step reasoning chain along with an answer, but again with no claim that the chain reflects the model's internal knowledge nor is valid reasoning (Subramanian et al., 2020).

To add semantics to generations, several systems have used self-querying to verify that generations reflect model-believed facts (by self-querying "Is p true?") (e.g., Kassner et al., 2021; Jung et al., 2022), or model-believed rules (by self-querying "Does p imply q?") (e.g., Tafjord et al., 2022). We build on these to construct a **belief graph**, namely a network of model-believed facts and their inferential relationships, which can then be reflected on.

**Beliefs:** We refer to the model's factual opinions as "beliefs" rather than "knowledge" because those opinions may be wrong. In general, an agent

can be said to believe p if it acts as if p was true (Schwitzgebel, 2019). Following Kassner et al. (2021) and Richardson et al. (2022), we take a simple, syntactic operationalization of this, namely the agent answers "yes" to the question "p?", but also note that more semantic versions could be used, e.g., the agent also answers "yes" to paraphrases and implications of p.

**Reducing Inconsistency:** LLMs are known to be inconsistent in their answers (Ettinger, 2020; Kassner and Schütze, 2020; Davison et al., 2019; Ravichander et al., 2020; Elazar et al., 2021; Subramanian et al., 2020; Gu et al., 2023), and several recent works have used constraint reasoners to identify and reduce inconsistency. BeliefBank used a MaxSAT solver to resolve inconsistencies between model beliefs, but required a hand-provided set of constraint rules (Kassner et al., 2021). ConCoRD (Mitchell et al., 2022) similarly used MaxSAT to ensure model answers were consistent with NLI-derived entailment constraints between them, but did not introduce additional model-believed facts and rules. Maieutic Prompting (Jung et al., 2022) also used MaxSAT to resolve inconsistencies between facts in prompt-induced explanation chains. However, those chains were not validated as reflecting model-believed constraint rules[2], and did not support conjunction. REFLEX extends these reasoning chains to provide a full semantic account of how answers are supported by the model's internal knowledge. Additionally, it performs joint reasoning across answer candidates and operates at a much larger scale (e.g., over 350 nodes on average for each question) and with a variety of constraint types.

## 3 REFLEX: Our Approach

### 3.1 Belief Graphs

Our belief graphs are defined over a set of natural language true/false *statements* and represent a set of *rules* that constrain the truth values of these statements. We refer to statements that are factually true in the world as *facts*. The truth value assigned by a model $M$ to a statement is referred to as $M$'s *belief* in that statement (cf. Footnote 1). A model's internal beliefs may not always align

with facts. Our goal is to extract a model's initial beliefs about statements inferentially related to all top-level hypotheses of interest, and perform reasoning to update these beliefs so as to make them more consistent with respect to the rules, and ideally also factually more accurate.

A belief graph is a type of *factor graph* commonly used in the probabilistic inference literature (Loeliger, 2004). Formally, it is defined as an undirected graph $G = (N, E)$ with nodes $N$ and edges $E$. Nodes are of two types: A *statement node* (referred to as a "variable node" in a factor graph) is a triple $(s, l, c_s)$ containing a natural language statement $s$, an associated value $l \in \{T, F\}$ initially denoting $M$'s belief that $s$ is true or false, and a confidence $c_s \in [0, 1]$ denoting a confidence in that label. A *rule node* (referred to as a "factor node" in a factor graph) is a pair $(r, c_r)$ denoting a disjunctive rule or constraint over statements, with confidence $c_r$. It takes the form $r = (-s_1 \vee \ldots \vee -s_\ell \vee s_{\ell+1} \vee \ldots \vee s_k)$. For ease of interpretation, we view this constraint as $r = p \rightarrow h$ where $p = s_1 \wedge \ldots \wedge s_\ell$ is a conjunctive premise and $h = s_{\ell+1} \vee \ldots \vee s_k$ is a disjunctive hypothesis. The rule says that if $p$ is true, so must be $h$; and the contrapositive of this.

Edges $E$ connect rule nodes to the statements they constrain, denoting their dependence. For legibility, we draw edges directionally to depict the way the rule reads: the statements in $p$ point to $r$, which in turn points to $h$. Mathematically, the influence is bidirectional and the depicted directionality is irrelevant during reasoning (Section 3.3), just as in a standard factor graph.

We adopt the standard probabilistic semantics of factor graphs, thereby associating a belief graph with a well-defined probability distribution over any set of statement beliefs. For a **statement node** $(s, l, c_s)$, the cost $cost_s$ for setting it to $l$ is 0, and that for setting it against $l$ is $c_s$; the corresponding *weight* of this node is $w_s = \exp(-cost_s)$. Costs and weights for a **rule node** $(r, c_r)$ are defined similarly, based on whether the beliefs satisfy $r$ or not. Finally, the overall weight of a T/F assignment to all statements is $\prod_s w_s \cdot \prod_r w_r$, which, when normalized by the total weight across all possible assignments, yields a probability distribution over such assignments. We will be interested in finding the *most consistent set of beliefs*, i.e., a T/F assignment to statements with the minimum overall weight, which is equivalent to minimizing

---

[2]REFLEX checks whether both the statements $s_i$, and the rules $(s_i \rightarrow h)$, are believed by the model via self-querying, e.g., by asking "Does $s_i \rightarrow h$?", and also scores the strength of those beliefs. In maieutic prompting, the generated rules are not checked against the model, resulting in rules that the model itself may not believe, if queried about them.

$\sum_s cost_s + \sum_r cost_r$. This is referred to as the MPE (most probable explanation) problem in the graphical models literature, which we later solve exactly using a MaxSAT constraint solver based on a standard translation of MPE into weighted MaxSAT (Park, 2002; Sang et al., 2007).

## 3.2 Constructing Belief Graphs

Given an initial node (statement) $s$, a belief graph $G$ is produced by a backward-chaining process described below, in which $G$ is recursively expanded to add statements that together may entail $s$.

### 3.2.1 Basic Operations

Let $h$ denote a hypothesis (language statement $s$) of interest and $p$ a premise—a set of statements $\{s_1, \ldots, s_n\}$ that together *may* entail $h$. Given these, there are **three basic operations** required to generate belief graphs:

**1.** $h \Rightarrow p$: Given $h$, *generate* a $p$ that may entail $h$.

**2.** $s \Rightarrow (l, c_s)$: Given a statement $s$, output a true/false value $l$ and a confidence in the belief that $s$ has truth value $l$ (as assessed via yes/no question-answering).

**3.** $(p, h) \Rightarrow c_r$: Given $p$ and $h$, output a confidence that the candidate rule $r = p \rightarrow h$ holds.

The most important of these is the first operation, in which the model self-generates conjunctive rules concluding $h$ (i.e., reason $p$ for believing $h$), thus adding new nodes to the graph.

There are several ways of implementing these basic functions, and our algorithm is agnostic to the method used. In our work here, we use Entailer, an off-the-shelf T5-11B trained model with these functionalities (Tafjord et al., 2022). Further, since the raw score produced by the model tends to be skewed towards 0 or 1, when computing $c_s$ and $c_r$ in practice, we re-scale the raw model score using a set of hyperparameters (cf. Appendix B).

One may use alternative ways to implement these operators, such as chain-of-thought prompting a model like GPT3 (Wei et al., 2022) or Chat-GPT (OpenAI, 2022). For example, to generate a rule concluding a hypothesis $h$ such as "Plants require CO2 to make their food.", the model could be prompted with $h$ followed by "Explain the last statement with a 2-step reasoning chain.", the numbered generations forming the premise $p$. Similarly, generated statements and rules can be validated as reflecting the model's beliefs by self-querying ("Is $s$ true?", "Does $p$ imply $h$?"), and then using the generated yes/no answer token probabilities as the

---

**Algorithm 1** The recursive algorithm for constructing a belief graph of max depth $d_{\max}$ for a hypothesis set $\mathcal{H}$. The subroutine EXTEND-GRAPH takes a partial graph $G$ as an input and extends it in place with one statement and its subgraph.

```
 1: procedure GENERATE-GRAPH(hypotheses H, max
       depth d_max):
 2:     let G = empty graph
 3:     foreach h ∈ H
 4:         call EXTEND-GRAPH(h, 0, d_max, G)
 5:     add MC rule node (⋁_{h∈H} h, ∞) to G
 6:     foreach pair (h_i, h_j) of hypotheses in H
 7:         add MC rule node (¬h_i ∨ ¬h_j, c_mc) to G
 8:     return G

 9: procedure EXTEND-GRAPH(statement s, current depth
       d, max depth d_max, partial graph G):
10:     call operator s ⇒ (l, c_s) to score statement s
11:     add statement node (s, l, c_s) to G
12:     gen. the negation sentence negs = neg(s)
13:     add rule node (XOR(s, negs), c_xor) to G
14:     call EXTEND-GRAPH(negs, d + 1, d_max, G)
15:     if d < d_max do:
16:         let h = s
17:         call operator h ⇒ p to generate p
18:         call operator (p, h) ⇒ c_r to score rule p → h
19:         add rule node (p → h, c_r) to G
20:         foreach s_i ∈ p
21:             call EXTEND-GRAPH(s_i, d + 1, d_max, G)
```

model's confidence (Kadavath et al., 2022).

### 3.2.2 Initial Hypothesis Generation

Given a question, we first generate a set $\mathcal{H}$ of hypothesis sentences (e.g., "Is the sky (A) blue (B) yellow" $\rightarrow$ { $h_1$ = "The sky is blue.", $h_2$ = "The sky is yellow.").[3] An $N$-way multiple choice question yields $N$ hypotheses in $\mathcal{H}$. A true/false question yields 2 hypotheses. To handle open-ended questions, candidate answers can be generated, e.g., using nucleus sampling (Holtzman et al., 2019).

### 3.2.3 Belief Graph Generation

The belief graph generation process is shown in Algorithm 1. An example of (part of) a generated belief graph is shown in Figure 2.

Given a set $\mathcal{H}$ of hypotheses, we generate a single belief graph $G$ by using our basic operations (Section 3.2.1) to recursively generate rules that conclude each $h_i \in \mathcal{H}$ up to a fixed maximum depth $d_{\max}$. (Each original $h_i$ is at depth $d = 0$.)

For each statement $s$, we also generate nodes $negs$ (and their recursive subgraphs) expressing its negation, e.g., "The sky is not blue." from "The

---

[3]Conversion of a QA pair to a declarative hypothesis D uses a custom T5-11B model trained on the QA2D dataset (Demszky et al., 2018).

sky is blue.".[4] Each pair $s$ and $negs$ is connected with an XOR rule, indicating a (soft) preference for setting exactly one of them to true; this is represented as two disjunctive constraints $(s \vee negs)$ and $(-s \vee -negs)$ whose weight $c_{xor}$ is a fixed hyperparameter. Lastly, we add a multiple-choice (MC) constraint which has two parts: a hard constraint (with infinite cost) that at least one hypothesis must be chosen, and a soft constraint[5] that no more than one should be chosen. The soft constraint is associated with a fixed hyperparameter weight $c_{mc}$.

### 3.3 Reasoning Over Belief Graphs

Belief graphs provide a window into the model's beliefs about some of the relevant statements and their (believed) inferential relationships to candidate answers to a question. As others have shown (Kassner et al., 2021; Mitchell et al., 2022), such beliefs can be inconsistent, and materializing those inconsistencies provides one the opportunity to remove or reduce them.

In a similar vein, and as discussed in Section 3.1, REFLEX performs inference over belief graphs in order to compute an updated set of beliefs that is as consistent as possible with the rules. To this end, it converts belief graphs into an equivalent weighted MaxSAT problem and uses an off-the-shelf MaxSAT solver (RC2, (Ignatiev, 2019)) to compute the optimal flips of initial true/false beliefs that minimize global inconsistency. It then discards all rules that are in conflict with the updated statement beliefs, obtaining a smaller, updated belief graph. This **smaller belief graph produced by REFLEX is self-consistent** and provides inferential support for the top-level hypotheses.

### 3.4 Generating Faithful Explanations

Notably, the smaller updated belief graph produced by REFLEX provides a **faithful** explanation of the answer it predicts, in the sense that it accurately represents the reasoning process behind *the overall system's* prediction (Lyu et al., 2022). This is true as the MaxSAT reasoning process results precisely in a self-consistent set of beliefs from which REFLEX determines whether to believe a candidate answer or not, and produces its final prediction based on this (rather than on the raw LLM output alone; note that we do not make any claims about

how the internal reasoning of the LLM component operates.) Thus, REFLEX provides the user with an interpretable reasoning trace, allowing the user to understand how it derived the answer from more rudimentary facts (Subramanian et al., 2020).

We note that the original belief graph (before reasoning) may reveal that the model's original explanation is, in fact, *not* faithful to its own beliefs. For example, in Figure 2, the model believes statements 6, 7, and that 6 & 7 entail 2, but does not believe 2 (colored grey). Thus, the global reasoning layer of REFLEX plays a critical role in arriving at faithful explanations.

## 4 Experiments and Results

The goal of our experiments is to evaluate the extent to which our overall system, namely an LLM plus a self-reflecting, rational layer, helps expose and resolve inconsistencies in the LLM's beliefs without harming accuracy. Importantly, REFLEX is evaluated in a *zero-shot* setting, without relying on training instances of the target datasets.

**Datasets.** We use the test partitions of three existing multiple-choice datasets: EntailmentBank (Dalvi et al., 2021), OBQA (Mihaylov et al., 2018), and QuaRTz (Tafjord et al., 2019). We chose our datasets as they contain inferentially rich questions (typically) requiring reasoning. The partitions contain 339, 500, and 784 examples, respectively.

**Models.** The **baseline LLM** we use is an LLM that has been trained to perform QA and also supports the basic operations discussed in Sec. 3.2.1, enabling us to assess how much it can be improved by adding a REFLEX layer. To this end, we use a publicly available, frozen, off-the-shelf T5-11B LLM called Entailer (Tafjord et al., 2022). To answer an MC question with this LLM, we score each answer hypothesis ($c_s$, Section 3.2.1) and select the one with the highest truth confidence. If Entailer assigns false values to all answer choices, we select the hypothesis with the lowest false confidence.

**REFLEX** then adds a rational layer to this LLM, creating a new system that is also able to self-reflect and modify its beliefs. To ensure the different belief graph scores in REFLEX are appropriately calibrated, we use nine hyperparameters, tuned once on the dev partition of EntailmentBank (Dalvi et al., 2021) and then kept fixed for all experiments. Details are in Appendix B. Note the LLM itself remains frozen, with belief revision occurring in the

---

[4] We use a simple, custom-built utility for this, namely a T5-base model trained on 9k Turk-generated examples.

[5] soft, to allow for cases with multiple valid answers, e.g., open-ended questions or those asking for the best answer.

rational (belief graph) layer above it.

**Metrics.** For measuring **self-consistency**, we follow Li et al. (2019) and report the *conditional constraint violation* ($\tau$) metric, defined as follows: the fraction of rules whose *premises p* are believed true, but whose *hypothesis h* is not. In other words, over all rules of the form $p \rightarrow h$, $\tau$ is:

$$\tau = \frac{|\{p \rightarrow h \mid p = \text{T}, h = \text{F}\}|}{|\{p \rightarrow h \mid p = \text{T}\}|}$$

where $s = T$ denotes the system believes statement $s$ to be true (similarly for $s = F$). The numerator of $\tau$ thus captures the number of constraints the system *violates*. The denominator captures the number of *applicable* constraints. We then report the following metric: **consistency = 1 - $\tau$**.

For **QA performance**, we report standard **multiple-choice accuracy**: 1 point for predicting the correct answer, $1/N$ points for predicting $N$ answers including the correct one, $1/k$ points for no prediction ($k$ = # answer options), 0 otherwise.

### 4.1 Results

**Consistency.** Table 1 shows consistency results on the test partitions of our datasets. We observe **significant consistency gains** (by 8%-11% absolute), showing REFLEX's effectiveness at creating a consistent belief network within the overall system.

| System | Entail-mentBank | OBQA | Quartz |
|---|---|---|---|
| LLM | 87.0 | 88.2 | 85.7 |
| LLM + rational layer (REFLEX) | **96.1** | **95.9** | **96.6** |

Table 1: **Consistency:** By adding a rational layer to the baseline LLM, REFLEX significantly improves consistency among beliefs by resolving uncovered conflicts.

**Accuracy.** Table 2 shows overall performance on our three datasets (test partitions). As can be seen, we observe stable accuracy, as well as the answers now being faithful to the reasoning chains in the belief graph. This is significant, as it allows users to understand how answers follow from system beliefs (and in cases where an LLM belief was flipped, why that belief is untenable in the broader system).

**Ablations.** To study the impact of the three different types of rules on consistency improvement, we using the EntilmentBank dataset (dev partition).

| System | Entail-mentBank | OBQA | Quartz |
|---|---|---|---|
| LLM | 79.4 | 74.0 | 80.2 |
| LLM + rational layer (REFLEX) | 79.9 | 75.0 | 80.0 |

Table 2: **QA accuracy:** REFLEX's belief revision in the rational layer preserves overall QA accuracy.

To do this, given the belief graph for a question, we mask out (separately, rather than cumulatively) each type of rule in turn when providing the graph to the MaxSAT solver. We then run the constraint solver and measure the resulting self-consistency of beliefs on the original graph.

| System | EntailmentBank |
|---|---|
| REFLEX (our system): | 96.1 |
| - without $p \rightarrow h$ rules | 93.8 |
| - without XOR rules | 90.4 |
| - without MC rule | 95.8 |

Table 3: **Consistency:** Ablations on EntailmentBank (Dev) suggest that all three types of rules contribute to improving self-consistency.

The results are shown in Table 3 (the MC rule is the constraint that exactly one multiple-choice option should be chosen, Section 3.2.3). The results indicate that all three types of rules contribute to the system's consistency improvements.

### 4.2 Success Analysis

We identify three classes of successful reasoning by the constraint reasoner: (a) latent model beliefs correct an initially wrong answer (Figure 3); (b) the system corrects an initially erroneous, latent model belief (Figure 4); and (c) strong model beliefs identify and reject a bad rule (Figure 5). These types of system corrections help to improve accuracy and produce answers supported by valid chains of reasoning, allowing users insight into why an answer follows from the model's knowledge.

### 4.3 Failure Analysis

Reasoning can also make mistakes. From a manual analysis of 50 random questions from EntailmentBank that REFLEX answered incorrectly, we identified five main causes of failure and their approximate frequency (**Note** that multiple categories can apply, hence total is $> 100\%$):

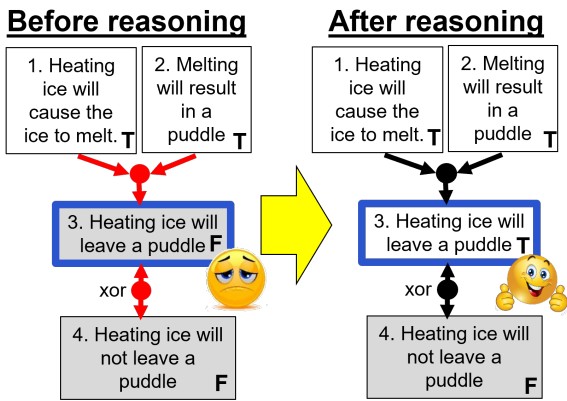

Figure 3: **Example of good reasoning:** The model's beliefs in 1 and 2, and the rule 1 & 2 → 3, as well as the xor constraint, causes it to (desirably) flip its belief in 3 from false (grey, before) to true (white, after).

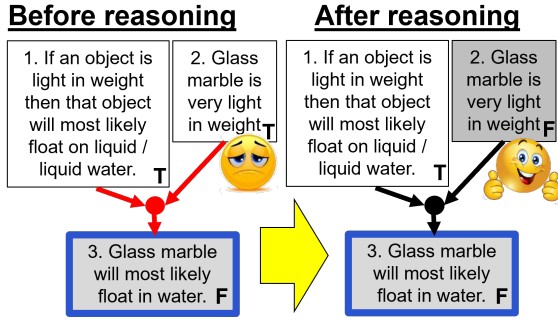

Figure 4: **Example of good reasoning:** Although the model correctly believes option (A) is false (grey, node 3), this answer conflicts with other beliefs (red). Reasoning leads the system to realize that its weakest belief (2) is actually false, correctly flipping its label from true (white) to false (grey, right side) restoring consistency.

**1. Missing Rules (≈30%):** In some cases, the system generates irrelevant rules but misses an important one needed to support the correct answer, resulting in incorrect conclusions. While somewhat subjective, this is a notable error category that we observe. For example for the question:

*A human cannot survive the loss of (A) The liver [correct] (B) A lung (C) A kidney*

the system incorrectly concludes (B) is true, ignoring the commonsense rule that with two lungs, a person can survive without one of them.

**2. Incorrect Beliefs (≈30%):** Sometimes the reasoner fails to correct incorrect model beliefs, either because the model's confidence is high or evidence against them is weak or missing. In the example shown in Figure 7, the model's strong, incorrect beliefs that "river deltas are reservoirs" and "reservoirs always provide freshwater" (untrue of oceans, say) causes it to incorrectly conclude that "deltas are freshwater reservoirs".

**3. Incorrect Rules (≈10%):** Rule generation can produce bad rules, e.g., in Figure 5), and in some cases the constraint reasoner fails to reject them if they are strongly believed. In particular, confusion or ambiguity over quantifiers can result in bad rules, e.g., (emphasis added) "*Some* animals catch their prey with trickery." & "A spider is a kind of animal." → "Spiders catch their prey with trickery.". Similarly the model generates the fallacy: "Some people don't mind not moving for an hour" & "breathing is a kind of movement" → "Some

people don't mind not breathing for an hour."

**4. Ambiguous Statements, Unexpected Reasoning (≈10%):** A common cause of error is the surprising ambiguity of belief statements, which can often be read in multiple ways. In several cases, the model adopts a valid but unexpected interpretation, resulting in "errors" compared to the gold answer label. For example, in Figure 6, the model takes the word "always" in a literal sense ("glaciers will not *always* be there"), resulting in an answer that differs from the gold label. Developing ways to attach context to these statements to help disambiguate them would help alleviate such errors.

**5. Multiple Valid Answers (≈10%):** A final cause of "error" - at least with respect to the gold label - is that multiple answers may be valid, and

What can build something over millions of years? (A) a river *[correct]* (B) a person (C) society (D) dinosaurs

Figure 5: **Example of good reasoning:** Here the reasoner (desirably) chooses to reject the violated (bad) rule rather than flip a belief, as the minimum cost way to restore consistency.

Which type of water reservoir could always provide freshwater? (A) river deltas (B) mountain glaciers *[correct]* (C) tropical seas

**Before reasoning**

1. Mountain glaciers would not always be there **T**

2. Glaciers provide a source of water used for melting **T**

3. Mountain glaciers could not always provide freshwater **F**

xor

4. Mountain glaciers could always provide freshwater **T**

**After reasoning**

1. Mountain glaciers would not always be there **T**

2. Glaciers provide a source of water used for melting **T**

3. Mountain glaciers could not always provide freshwater **T**

xor

4. Mountain glaciers could always provide freshwater **F**

Figure 6: **Unexpected reasoning:** Here the model unexpectedly pays particular attention to the world "always". Because it strongly believes that glaciers will not *always* be there (1, white), the system prefers to flip its beliefs in 3 and 4, rather than flipping 1, thus rejecting answer option B (arguably correctly).

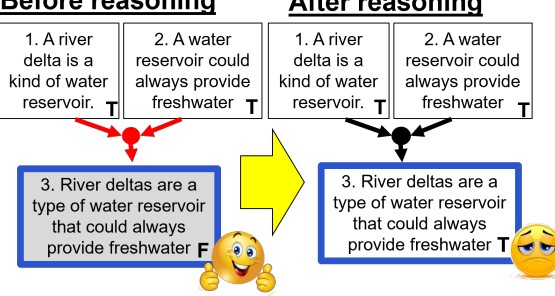

Which type of water reservoir could always provide freshwater? (A) river deltas (B) mountain glaciers *[correct]* (C) tropical seas

**Before reasoning**

1. A river delta is a kind of water reservoir. **T**

2. A water reservoir could always provide freshwater **T**

3. River deltas are a type of water reservoir that could always provide freshwater **F**

**After reasoning**

1. A river delta is a kind of water reservoir. **T**

2. A water reservoir could always provide freshwater **T**

3. River deltas are a type of water reservoir that could always provide freshwater **T**

Figure 7: **Failure due to bad beliefs:** The model strongly believes both 1 and 2 (although both are factually incorrect), here causing 3's label to undesirably flip from false (correct) to true (incorrect).

the question is asking for the **best** answer; eg. for "What could fill a beach ball? (A) Oxygen (B) Water ...", A is labeled correct, while B is also a valid answer. REFLEX (desirably) finds valid reasoning chains for both, but the notion of highest-scoring proof does not fully correlate with the notion of "best answer" intended by the question author.

## 5 Future Work

There are several impactful ways this work could be further extended. First, incorporating the question's *context* in the belief statements in our rational layer could make the semantics of the beliefs more precise, thus avoiding potential ambiguity in their truth value. Second, one could use the belief graph itself to identify the key reasoning pieces that the LLM is most uncertain about. This could then guide a *human-in-the-loop* mechanism to correct or validate uncertain pieces via user interaction. Third, maintaining a *persistent belief graph* over multiple questions could help make the system more consistent across questions. This, in turn, would make a user's conversational experience with the system more coherent in a longer dialog setting. Lastly, after resolving inconsistencies in the rational layer, we could consider *propagating information back to the LLM layer* in order to update it (via fine-tuning, model editing, memory-based architectures, etc.),

helping avoid similar inconsistencies in the future.

## 6 Conclusion

While LLMs perform well, the interdependencies between their answers and their other beliefs is opaque, and may even be in conflict. This lack of interpretability is a significant impediment to widespread use of LLMs. To reduce this opacity, and reduce these conflicts, we have proposed REFLEX, a new system architecture in which an explicit, interpretable representation of beliefs - the **belief graph** - is added as a **rational layer** above the LLM. This layer providing a window into system beliefs, and allows latent inconsistencies in the LLM alone to reasoned about and repaired. Our implementation shows that belief consistency of the overall system is significantly improved, without harming answer accuracy, resulting in answers supported by interpretable chains of reasoning drawn from a more consistent belief system. This new architecture is an important step towards improving confidence in system behavior, and towards trustable deployment of LLMs in practical applications.

## Limitations

We have shown how an LLM can be extended with a self-reflective component, allowing latent model knowledge to be made explicit in the form of a **belief graph**, providing a window into the model's system of beliefs. While exciting, there are several limitations with the current work and opportunities for the future.

First, the reasoning component in the rational

layer can make mistakes, resulting in the overall system rejecting true statements or accepting false ones. A detailed analysis and classification of these failure modes was presented in Section 4.3.

Second, for our experiments, we used the T5-11B based Entailer system as the baseline LLM. While there is every reason to expect our proposed architecture to be effective in reducing inconsistency with newer and larger LLMs such as ChatGPT and LLaMA, this is still to be evaluated. Doing so would require implementing the basic operations needed to construct belief graphs (Section 3.2.1) using instruction prompting and in-context learning. Other work has demonstrated such implementations (e.g., Wei et al., 2022; Jiang et al., 2020), making the outlook promising, but indeed their combination still needs to be demonstrated at scale in an architecture like REFLEX.

Lastly, we found consistency-minimized belief graphs to be highly valuable in understanding the system's successes and failures. We expect these graphs to be a valuable starting point for providing explanations and gaining a user's trust in the system. However, we have not conducted a formal user study to measure this.

## Ethics Statement

Like any other project using LLMs, despite the best intentions there is a risk of the model producing biased or offensive statements as part of its explanations, and thus must be used with care and appropriate guards and warnings.

## Acknowledgements

This research was made possible, in part, by funding from Open Philanthropy, the European Research Council (#740516) and by the German Federal Ministry of Education and Research (BMBF) under Grant No. 01IS18036A. We also thank Google for providing the TPUs for conducting experiments. Finally, we are grateful for the valuable feedback from the anonymous reviewers.

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

## A  Additional Results

We report results on the dev set of the Entailment-Bank dataset in Table A1.

| System | EntailmentBank (dev) | |
|---|---|---|
| | Consistency | Accuracy |
| LLM | 87.5 | 78.6 |
| LLM + rational layer (REFLEX) | **96.1** | **81.8** |

Table A1: Results on EntailmentBank (dev), used to tune the system's hyperparameters.

## B  Hyperparameters and Runtime

MaxSAT finds the optimal assignment of true/false labels on statement nodes that minimizes the total penalty of constraint violations. If the true/false label on a statement node is flipped, then the penalty is the model confidence $c_s$ in the original label. Similarly if a rule (constraint) is violated by the true/false labels on its associated statements, then the penalty is the model confidence $c_r$ in that rule.

We set a number of hyperparameters to ensure that the various sources of confidence are appropriately balanced, and tune these on a development set (EntailmentBank (dev) which is separate from our test sets). We use the same set of hyperparameters for all test sets.

1. As raw model confidences $c_s$ are highly skewed towards 0 and 1, we re-calibrate these with $e^{k.(c_s-1)}$, where k is a fixed hyperparameter. Note, that for the MC and XOR rule, the raw input score $s$ is 1.0.

2. We calibrate rule confidences in the same way as we calibrate belief confidences but use separate calibration parameters different types of rules namely:

   - Entailer rules $p \rightarrow h$
   - XOR rules
   - MC rules

   i.e., the raw rule score $c$ is re-calibrated to confidence $e^{k_{type}.(c-1)}$ where $k_{type}$ is the respective hyperparameter per rule type.

3. We set three hyperparameters tuning the respective importance of the three different types of rules. Therefore, the final rule score is computed by $c = t_{type} * e^{k_{type}.(c-1)}$ where $t_{type}$ is the respective hyperparameter constant per rule type.

4. For xor rules between statements $s_i$ and $negs_i$,

| Hyperparameter | Value |
|---|---|
| $k$ | 9 |
| $k_{\text{entailer}}$ | 36 |
| $k_{\text{xor}}$ | 30 |
| $k_{\text{mc}}$ | 9 |
| $t_{\text{entailer}}$ | 1.02 |
| $t_{\text{xor}}$ | 1.1 |
| $t_{\text{mc}}$ | 0.98 |
| $m_{\text{xor}}$ | 0.3 |
| $d_{\max}$ | 5 |

Table B1: Hyperparameters.

we remove (ignore) those where there is significant uncertainty, namely where $|score(s_i) - score(negs_i)| \leq m_{xor}$, where $m_{xor}$ is a tuned hyperparmeter.

5. Additionally, we tune a damping parameter that downscales rules on the boundary of the graph. Belief nodes involved in these rules are not supported by any premises and should therefore have less influence than rules with strong support.

6. Finally, we tune the maximum depth $d_{\max}$ of the belief graph.

The performance on this dev set partition is shown in Table A1 and the hyperparameter values are shown in Table B1.

The runtime for MaxSAT constraint solving is fast (<1 millisecond per question). However, constructing the belief graph is computationally intensive: Each call to expand or score a node takes ∼2 seconds, and our graphs typically contain ∼600 nodes, so if these calls were maximally parallelized, with each step growing the graph one level deeper, the runtime would be the maximum graph depth (5) x 2 seconds = ∼10 seconds total (or several minutes if a naive sequential implementation were used).

