# OpenReview forum: "Language Models with Rationality"
_EMNLP/2023/Conference — EMNLP 2023 Main_

### Official Review · Reviewer_mJrZ · 2023-07-31

**Soundness:** 4

**Excitement:**

4: Strong: This paper deepens the understanding of some phenomenon or lowers the barriers to an existing research direction.

**Paper Topic And Main Contributions:**

Research question: This paper works on materializing language models' internal knowledge to enhance the interpretability of models in QA tasks.

Contribution:
1. It introduces a rational layer (which is in fact separated from the language model) which consists of belief graphs i.e. explicit representation of system beliefs and their logical and inferential relationship. This layer serves as a medium to interpret the model's output and allows reasoning capability of the model to be improved.
2. Implementation of the layer
3. Experiments (in the zero-shot manner) on three datasets show that consistency of the overall system's network of beliefs can be significantly improved without harming answer accuracy.



**Questions For The Authors:**

1. Line 198-200: Why Maieutic's explanation chain are not validated as reﬂecting model-believed constraint rules (or not able to expand to such a belief graph that facilitates that)?
2. According to my knowledge, the full test set size of OBQA is 500 (ref https://allenai.org/data/open-book-qa), of QuaRTz is 784 (ref https://allenai.org/data/quartz). Why does this paper has such a numerical discrepancy?
3. In Section 4.3, among 50 random questions, can you give the exact number of mistakes (I assumes it is equivalent to incorrect conclusions or incorrect answers, please clarify if my assumption is wrong) that the model makes? Also the exact number of samples in each failure categories. In those failure categories, which one surely entails incorrect answer for the test questions (e.g questions from EntailmentBank)?
4. I believe that when we represent the reasoning chain of LMs decently, the next step is to use it to improve the reasoning capability. Since the naive decision made by the optimized belief graph does not work, do you have any general idea to use the belief graph in a different way to improve model reasoning capability? e.g. propagating the updated belief graph back to the model during inference time, or design a pretraining task to improve model consistency?

**Reasons To Accept:**

1. Belief graph advances the model internal knowledge's materialization by facilitating richer logical relations (a.k.a rule node) and on-the-fly context-dependent belief generation. Like some other previous work, the graph serves as proxy to interpret model's output, as well as allows to externally fix the inconsistency among those belief. This will benefit a lot of reasoning tasks.
2. The method is clearly described and illustrated, including the construction of the belief graph from the initial state and how to optimize it.
3. The paper demonstrates thorough quantitative and qualitative analyses, especially the case of good reasoning and failure, and the relationship of model false belief and failure in the belief graph optimization stage.


**Reasons To Reject:**

1. The belief graph to some extent are problematic. [1] suggests that language models' generated text do not always reflex what they internally "think", thus the belief graph may not faithfully represent the rationale for the models' output (yet it is a common argument against all explitcit interpretation methods).

2. Although the paper does not claim that the belief graph improves the reasoning capablity, I believe that the authors did attempt to do so by determining the final answer of LMs using the optimize belief graph. Given that situation, experimenting on data sets which are different from that in previous works ([2], [3]) hinders the comparison between different methods.

[1] Language Models Don't Always Say What They Think: Unfaithful Explanations in Chain-of-Thought Prompting, Turpin et. al. (2023)

[2] Enhancing Self-Consistency and Performance of Pre-Trained Language Models through Natural Language Inference, Mitchell. et al. (2022)

[3] Maieutic Prompting: Logically Consistent Reasoning with Recursive Explanations, Jung et. al. (2022)


**Reproducibility:**

3: Could reproduce the results with some difficulty. The settings of parameters are underspecified or subjectively determined; the training/evaluation data are not widely available.

**Reviewer Confidence:**

4: Quite sure. I tried to check the important points carefully. It's unlikely, though conceivable, that I missed something that should affect my ratings.

---

> ### Author Rebuttal · Authors · 2023-08-28
>
> Thanks for your thorough review and encouragement!
>
> *>The belief graph to some extent are problematic. [1] suggests that language models' generated text do not always reflex what they internally "think", thus the belief graph may not faithfully represent the rationale for the models' output (yet it is a common argument against all explitcit interpretation methods).
>
> Yes, care is needed to interpret the graph. Importantly, you are correct that chains in the graph do not necessarily represent how the model *actually* reasoned internally, and we make no such claims (Lines 385-7). Rather, the (initial) graph is a snapshot of the model's beliefs and their (model-believed) inferential relationships, hence an explanation from the (initial) graph denotes one model-believed chain of reasoning to an answer, even if it actually arrived at that answer via a different mechanism. We think of it as an orthogonal yet still model-believed “proof” of an answer.
>
>
> *>Although the paper does not claim that the belief graph improves the reasoning capability, I believe that the authors did attempt to do so by determining the final answer of LMs using the optimize belief graph. Given that situation, experimenting on data sets which are different from that in previous works ([2], [3]) hinders the comparison between different methods.
>
> We chose our datasets as they contain inferentially rich questions (typically) requiring reasoning, the focus of our work. You are correct that we could add datasets from [2] and [3], although it's not clear how to then draw comparisons: Specifically, we can't simply compare accuracies with published results as the underlying models differ; and as [2] and [3] have restricted expressivities (e.g., they don't represent rules with multiple premises), it is non-trivial how one might generate a comparable belief graph with them to fairly compare consistencies. However, there are qualitative comparisons that can be drawn, which we attempted in the Related Work.
>
> *>1. Line 198-200: Why Maieutic's explanation chain are not validated as reﬂecting model-believed constraint rules (or not able to expand to such a belief graph that facilitates that)?
>
> Yes: REFLEX checks whether both (a) the statements s_i, and (b) the rules (s_i -> h), are believed by the model via self-querying, e.g., by asking “Does s_i -> h?”, and also scores the strength of those belief. In maieutic prompting, only (a) is done - the generated rules are not checked against the model, hence maieutic prompting may generate a rule that it itself may not believe, if it queried itself about it. (This self-check could be easily added to the next generation of maieutic prompting work).
>
> *>2. Exact number of mistakes …
>
> Thank you for catching this, it is a simple typo, we forgot to update the numbers. We worked with smaller subsets in the past but used the full test sets for this submission. The results were computed on the full test sets. We will correct this in future version.
>
> *>3. In Section 4.3, among 50 random questions, can you give the exact number of mistakes (I assumes it is equivalent to incorrect conclusions or incorrect answers, please clarify if my assumption is wrong) that the model makes? Also the exact number of samples in each failure categories. In those failure categories, which one surely entails incorrect answer for the test questions (e.g questions from EntailmentBank)?
>
> Yes, just to clarify: all 50 were cases sampled from when the system produced the incorrect answer, and our goal was to identify the primary cause(s) of failure overall. We will clarify this in the introduction to 4.3, our apologies.
> In the five failure categories, the counts respectively were: 16, 17, 4, 6, and 4, indicating that (1) failing to generate an important rule, and (2) having bad beliefs, were the primary causes of wrong answers.
>
> *>4.I believe that when we represent the reasoning chain of LMs decently, the next step is to use it to improve the reasoning capability. Since the naive decision made by the optimized belief graph does not work, do you have any general idea to use the belief graph in a different way to improve model reasoning capability? e.g. propagating the updated belief graph back to the model during inference time? or design a pretraining task to improve model consistency?
>
> Yes, we are completely on the same page here, this is a natural follow-up direction to look into. We have an elaborate error analysis examining why accuracy improvement is minimal, and attacking these could improve systematic reasoning accuracy. Your suggestions are right on: somehow propagating updated beliefs back into the model would be highly desirable (though current model editing techniques are not quite up to this), and similarly pretraining for consistency may improve the model's initial self-consistency (e.g., [Ribeiro et al., 2019]). Additional approaches we have considered include: identifying areas of maximal conflict in the belief graph and doing additional localized inference to materialize more knowledge to help resolve conflicts the right way; using external resources to help identify and correct bad model beliefs; or even include a user in the loop, e.g., using the belief graph to identify areas of model confusion, and leveraging that to ask the user judicious questions to resolve the conflict. These would all be great directions for future work.
>
> [Ribeiro et al., 2019] Are red roses red? ACL.

---

### Official Review · Reviewer_udQV · 2023-08-04

**Soundness:** 4

**Excitement:**

5: Transformative: This paper is likely to change its subfield or computational linguistics broadly. It should be considered for a best paper award. This paper changes the current understanding of some phenomenon, shows a widely held practice to be erroneous in someway, enables a promising direction of research for a (broad or narrow) topic, or creates an exciting new technique.

**Paper Topic And Main Contributions:**

This is an NLP experiment paper.  This work proposes a system called "REFLEX" that builds large (hundreds of nodes) belief graphs from language models, to support inferring whether the answer to a given question is correct.  The belief graphs include both facts relevant to inferring whether the answer to a question is correct, as well as rules.  Using a combination of querying the LLM for self-consistency, as well as GOFAI techniques (probabilistic reasoning graphs, constraint satistfaction solvers), the system is shown to generate substantially better belief graphs than a baseline system without this modification.


**Reasons To Accept:**

- Self-reflection has become common in code-generation models, for models (like ChatGPT) to debug their own code.  This paper essentially builds a system for doing something similar with graphs of an LLM's beliefs about the world, while applying a neurosymbolic approach (LLM + probabilistic graphs + constraint reasoner) to solve the task at scale ("350 nodes and 80 constraints per question").  This is a substantial and elegant contribution, making the knowledge an LLM uses to reason much less opaque.

- The system achieves a large (~10%) performance increase in the task of generating accurate belief graphs compared to previous models, on three benchmark datasets (EntailmentBank, OpenBookQA, and Quartz).

- Good literature review for the paper's size

- Clear exposition of the model

- Straightforward results, clear evaluation measurements, ablation testing.

- Detailed error analysis


**Reasons To Reject:**

- I don't see any reasons for rejecting this paper

**Reproducibility:**

4: Could mostly reproduce the results, but there may be some variation because of sample variance or minor variations in their interpretation of the protocol or method.

**Reviewer Confidence:**

4: Quite sure. I tried to check the important points carefully. It's unlikely, though conceivable, that I missed something that should affect my ratings.

**Typos Grammar Style And Presentation Improvements:**

- "eg" -> "e.g." (L164, L166)

---

> ### Author Rebuttal · Authors · 2023-08-28
>
> Thanks for your thorough review and encouragement! We really appreciate your crystal-clear summary and enthusiasm for the work!

---

### Official Review · Reviewer_jJr9 · 2023-08-05

**Soundness:** 4

**Excitement:**

4: Strong: This paper deepens the understanding of some phenomenon or lowers the barriers to an existing research direction.

**Paper Topic And Main Contributions:**

This paper proposes to extend a language model with a "reflection" system which queries the model for beliefs, assembles those beliefs into an implication graph, then harmonizes this graph via MaxSAT in order to improve the consistency and faithfulness of automated natural language reasoning.

The authors evaluate the effectiveness of their proposal using three multiple-choice question-answering datasets, assessing system performance in terms of both the consistency of explanations and multiple-choice accuracy. The results indicate that the proposed reflection harness drastically improves the overall system's consistency while maintaining the accuracy of the underlying model.

**Questions For The Authors:**

A: You mention that you use the model's confidence in answering "yes" to a particular statement to assess beliefs, and that a more "semantic" notion of belief would involve querying multiple paraphrases. Anecdotally, do you see evidence that there is substantial inconsistency in belief scores across paraphrases?

**Reasons To Accept:**

- Attempting to construct systems that provide *faithful* explanations while performing on par with less-structured competitors is usually difficult, so achieving the reported consistency gains without hurting system performance is impressive.

- While the question of whether models can be said to have "beliefs" at all is open to interpretation, the authors do a good job of making their working definitions explicit, and operationalizing them in such a way that the whole thing hangs together, i.e. the system's explanations are causally faithful, but it still makes sense to frame them as products of the model's "knowledge".

**Reasons To Reject:**

- The three datasets used for evaluation are all roughly aligned with the style of text the base model (Entailer) is tailored to. This raises the question of whether the approach is actually portable to other domains/LMs which haven't been explicitly tuned to present the functionalities listed in section 3.2.1. While the authors sketch a means of implementing these operations using few-shot capabilities, the broader usefulness of the contribution would be less questionable if evaluated in other domains.

**Reproducibility:**

4: Could mostly reproduce the results, but there may be some variation because of sample variance or minor variations in their interpretation of the protocol or method.

**Reviewer Confidence:**

4: Quite sure. I tried to check the important points carefully. It's unlikely, though conceivable, that I missed something that should affect my ratings.

**Typos Grammar Style And Presentation Improvements:**

Line 463: ... how answers [follow/derive/?] from...

Visually, figure 1 is a little cluttered/overwhelming. It might be more useful for readers if the LLM Layer was shown as a simple box, maybe with one or two arrows to belief nodes indicating that beliefs are extracted from the model (the current vertical yellow arrows are ambiguous).

---

> ### Author Rebuttal · Authors · 2023-08-28
>
> Thanks for your thorough review and encouragement!
>
> *>How portable is the approach to other LMs which haven't been explicitly tuned to present the functionalities listed in section 3.2.1?
>
> In principle if the required operations are implemented using few-shot capabilities, then the approach should port easily. Other work has demonstrated the individual components already exist: chain-of-thought systems demonstrate premise generation (h -> p) [Wei et al 2022]; and several methods exist for assigning confidences to LLM's answers about whether a fact/rule is true, e.g., [Jiang et al., 2021]. This makes the outlook promising, but indeed their combination still needs to be demonstrated at scale in an architecture like REFLEX.
>
> [Wei et al 2022] Chain of Thought prompting elicits reasoning in LLMs. NeurIPS.
> [Jiang et al., 2021] How can we know when LMs know? TACL.
>
> *>You mention that you use the model's confidence...to assess beliefs, and that a more "semantic" notion would involve querying multiple paraphrases. Anecdotally, do you see evidence that there is substantial inconsistency in belief scores across paraphrases?
>
> Good question: Anecdotally, for the model we used, the truth assignment seems very consistent among paraphrased statements, but the scores can still vary depending on phrasing. But we would expect high confidence scores to correlate with more consistency across paraphrases, and overall behavior is mainly affected by high-confidence scores.
>
> *>Figure 1 is a little cluttered/overwhelming
>
> Thanks for this feedback and useful suggestions for improving the figure! We will be sure to improve it in future versions!

---

### Meta-Review · Area_Chair_PanC · 2023-09-20

**Recommendation:** 5

**Metareview:**

This paper proposes to add a component to LLMs that constructs a "belief graph", repair inconsistencies, and produce faithful reasoning chains drawn from a belief system with significantly improved consistency. All reviewers praised the work, highlighting the provided consistency gains, exposition, and analyses. Some concerns were inherent to whether the methodology proposed in this paper could transfer to other domains, about the choice of the datasets, and whether the belief graphs faithfully reflect how the model operates internally. The authors provided clarifications in their rebuttal. Overall, the reviewers found the paper a significant contribution to NLP, which could open interesting research avenues. Two reviewers out of three recommend it for a best paper award.

---

### Decision · Program_Chairs · 2023-10-07

**Decision:**

Accept-Main

**Comment:**

This paper proposes to add a component to LLMs that constructs a "belief graph", repair inconsistencies, and produce faithful reasoning chains drawn from a belief system with significantly improved consistency. All reviewers praised the work, highlighting the provided consistency gains, exposition, and analyses. Some concerns were inherent to whether the methodology proposed in this paper could transfer to other domains, about the choice of the datasets, and whether the belief graphs faithfully reflect how the model operates internally. The authors provided clarifications in their rebuttal. Overall, the reviewers found the paper a significant contribution to NLP, which could open interesting research avenues. Two reviewers out of three recommend it for a best paper award.